# Evaluation of a multimodal pain therapy approach with relapse prophylaxis for back pain (MMS-RFP study): a study protocol for a cluster randomised controlled trial

Kathrin Krueger,[1,2] Julia Schmetsdorf [ID] ,[1,2] Maja Pavlovic,[3] Werner Runde,[4] Georg Zechel,[4] Norbert Hemken,[4] Christian Krauth[1,2]

KK and JS contributed equally.

KK and JS are joint first authors.

For numbered affiliations see end of article.

**Correspondence to**
Julia Schmetsdorf;
Schmetsdorf.Julia@mh-hannover.de

## ABSTRACT

**Introduction** The need for an interdisciplinary multimodal approach to the treatment of back pain has already been demonstrated by various studies. However, when considering the periods of incapacity to work in the longitudinal course after the multimodal pain therapy (MPT), limits in terms of its sustainable effect become clear. Patients who receive MPT subsequently return to standard outpatient care, which is associated with a risk of relapse. A 12-month relapse prophylaxis (RP) programme, intended to follow a 4-week MPT, was developed to help patients make the transition to health-conscious and physically active behaviour in everyday life and to identify and prevent impending relapses at an early stage. The evaluation, based on a cluster randomised controlled trial, seeks to provide information on the benefits of early and intensive RP as part of MPT, examine whether it is cost effective, reduces the days of incapacity to work and increases functional capacity, as well as to examine other parameters.

**Methods and analysis** The study population comprises members of a regional statutory health insurance fund in Germany, who are ≤62 years old, gainfully employed and have been incapacitated for work for at least 21 days due to a diagnosis of back pain. Over a recruitment period of 24 months, a maximum of 368 individuals can potentially be included in the MPT. The intervention group (IG) and control group (CG) will both receive MPT, after randomisation IG will receive RP and CG will receive no further therapy or support as part of the trial. The evaluation is carried out on the following levels: structural, process and results quality. Cost effectiveness is also assessed by means of a health economic evaluation. In addition to the collection of qualitative and quantitative primary data, claims data from the regional health insurance fund are also included in the analysis.

**Ethics and dissemination** This study has received approval by the ethics committee of the Hannover Medical School (reference number: 8548_BO_S_2019). The study results will be disseminated in national and international journals and conference presentations.

**Trial registration number** DRKS00017654.

## STRENGTHS AND LIMITATIONS OF THIS STUDY

⇒ A comprehensive evaluation will be provided regarding the ability of this new treatment approach in form of relapse prevention to treat back pain adequately and, furthermore, support patients in their behaviour and motivation to take care of their back health themselves before chronicity occurs.

⇒ The results will contribute a better understanding of whether the acceptance of the programme could be increased by involving and closely accompanying the patients and providing them with concrete training instructions, which highlights high demands on the treatment concept that are not covered by standard outpatient care.

⇒ The primary outcome is not collected via questionnaires but rather via claims data, so that an information bias can be excluded in this respect.

⇒ As a cluster-randomised study with monthly randomisation, a certain degree of contamination, due to personnel changes, for example, is unavoidable, which is a limitation of the methodology.

⇒ As a limitation for the primary outcome, it should be taken into account that the days of incapacity to work are based on incapacity certificates issued by the treating physician, which are usually only submitted to the health insurance funds for incapacity to work lasting 3 days or longer, but since a period of incapacity to work for more than 2 days is usually to be expected in the case of back pain, the impact is likely to be small.

## INTRODUCTION

Back pain has a high prevalence in today's society and is one of the most common complaints in the population. In addition to frequency of occurrence, the main criteria for the assessment of back pain are severity, extent of functional impairment and prognosis with regards to return to functional and thus work capacity. Data on the prevalence of back pain in Germany include results from

several regional and national surveys. The range of the prevalence lies between 30% and 70%, depending on the period considered (point prevalence, 7-day, 3-month and 1-year prevalence).[1] The lifetime prevalence for back pain in Germany ranges from 74% to 85%.[2] In terms of chronic back pain, the lifetime prevalence lies between 24% and 30%.[3] With regards to incapacity to work, there are various references in the literature.[2 4–6] Sixty eight per cent to 86% of patients with acute low back pain return to work within 1 month.[4] The recurrence rate of low back and neck pain is 47%–63%, the relapse rate into inability to work is 33%.[1 5 7] Relapse refers to the recurrence of back pain after temporary improvement. Musculoskeletal disorders are among the most expensive diseases in industrialised countries.[8] According to the 2008 Federal Health Report, the costs of non-specific low back pain amounted to €3.6 billion.[2 8] The majority of these costs are due to chronic low back pain and indirect costs resulting from the loss of patients' ability to work. Wenig *et al* estimated the average total costs of back pain per patient to be €1322 per year.[9 10] As a cause of premature retirement due to reduced earning capacity, musculoskeletal disorders have taken second place in recent years after mental illness and behavioural disorders.[2 8] A large percentage of the elderly population suffers from one or more musculoskeletal disorders. However, chronic back pain, reported by about a quarter of women and 17% of men in Germany, often affects younger people in working life.[11]

Back pain is usually treated in a monodisciplinary manner, although psychosocial risk factors in particular often hinder recovery and require a multimodal approach with close content-related therapy consultation.[12 13] Interdisciplinary multimodal treatment is not yet implemented in standard outpatient care.

Interdisciplinary multimodal pain therapy (MPT) describes the simultaneous, comprehensive treatment in which various somatic, physical and psychological interventions are integrated according to a predefined treatment plan with an identified therapeutic goal. The treatment is provided in small groups with a maximum of eight patients by a team of therapists consisting of physicians, psychologists or psychotherapists and other disciplines such as physiotherapists and others.[14] MPT is currently considered the gold standard in pain management and treatment for patients with chronic or potentially chronic back pain.[15] The efficacy of interdisciplinary MPT in the treatment of chronic or recurrent back pain with risk of chronification has already been proven in various studies.[8 13 14 16–20] The term recurrent describes pain episodes that reoccur after a symptom-free period of at least 6 months.[8] Chronification, moreover, describes the transition from temporary to permanent, or chronic, back pain. After successful MPT, patients usually return to the monodisciplinary treatment of their general practitioner or orthopaedist. As a result, the risk of recurrence remains with the end of MPT. The goal of this new approach to the treatment of back pain is to achieve sustainability of MPT by combining MPT with relapse prophylaxis (RP). Thus, what is learnt during multimodal treatment in terms of physical activity and lifestyle can be applied to everyday life in the long term.

The intervention examined in this study, the 12-month RP programme, immediately follows the 4-week MPT. Like the MPT, the RP will be performed by the Rehabilitation Center, Bad Zwischenahn, on an outpatient basis. The rehabilitation centre is completely independent of the evaluating institution. The RP aims not only to provide prophylaxis against chronifying back pain relapses but also to support life-integrated physical activity. RP is the all-important link between the curative MPT phase beforehand and the patient's physical activity afterwards. The patient will be assigned to one of three RP paths at the end of the MPT as part of an assessment during the interdisciplinary rounds and team meetings. The paths correspond to RP programmes with three different intensities (low, medium and high) for which the patient will be selected based on the severity of his or her complaints (see figure 1).

The low intensity level will include the opportunity for telephone consultation and the maintaining of an exercise diary as well as an obligatory telephone conversation after 6 weeks. After 6 months and again after 1 year, a 3-hour training session will take place at the Rehabilitation Center with performance tests, examination of the exercise diary and therapeutically supervised training by physiotherapists and orthopaedists. In addition, a psychological group session will be conducted by psychologists.

The medium intensity level will additionally include a training session after 3 months and another obligatory phone call in the 4th week and 8th week after the MPT.

In the case of high-intensity RP, a weekly training session (12 weeks of 90 min each) under therapeutic supervision will be scheduled directly after the MPT. Sixteen weeks and 20 weeks post MPT, obligatory telephone calls with the patient will be conducted. The follow-up training course will be supplemented by another session after 9 months. For patients with a particularly high need for follow-up treatment, a 1-week training course will be offered after 12 months.

The RP offers the advantage of facilitating the patient's transition into everyday life, accompanying and encouraging progress, as well as enabling early recognition and prevention of relapses. The further development of care thus consists of uninterrupted, seamless and long-term care of patients with back pain following MPT. Another goal is to motivate patients during both the MPT phase and the subsequent 12-month RP phase under multiprofessional guidance to adopt a lifelong active lifestyle. While developing good intentions seems easy, turning them into actions is a challenge. For many, the hurdles to a permanently active lifestyle in everyday life are high.[21] RP, therefore, serves primarily to provide professional support for the patient in the conscious, voluntary translation of goals and motives into results. For patients suffering from back pain, this can be regarded as a secondary prophylaxis

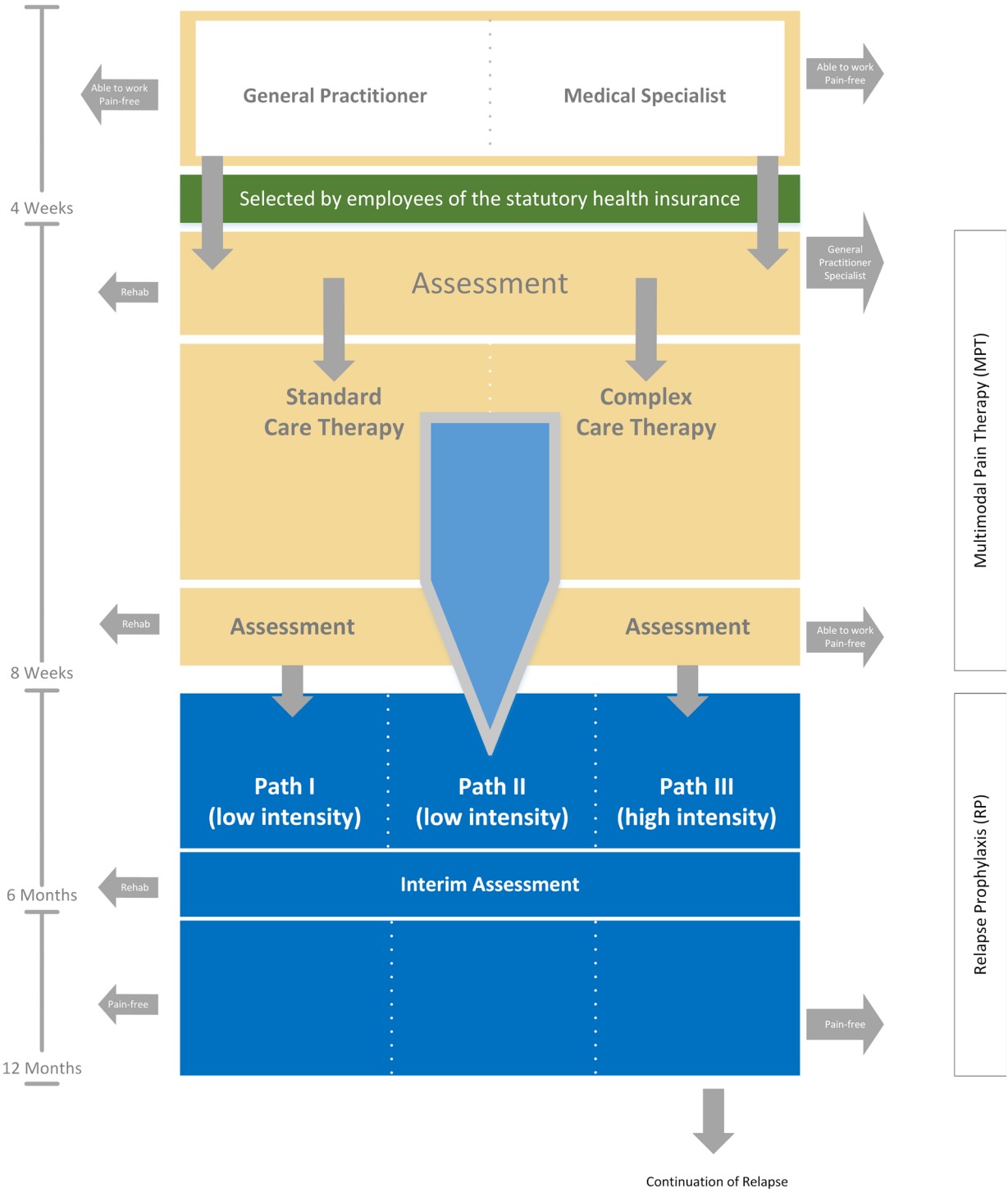

**Figure 1** Treatment algorithm.

against renewed pain, functional impairment and incapacity to work. In this respect, patients at risk of recurrent back pain are on the same level as patients with other chronic widespread diseases, such as diabetes or cardiovascular diseases, for which there has long been a consensus on appropriate prevention.

## Multimodal pain therapy

Employees of the AOK Lower Saxony will contact the potentially eligible insured and refer them to the Rehabilitation Center. Before starting MPT, a comprehensive initial examination will be performed by orthopaedists, physiotherapists and psychologists. The MPT programme intensity depends on the severity of the back pain. Patients in the standard MPT group will be treated for 3–4 hours per day, 4 days a week, while patients in the complex MPT will be treated for 5–7 hours per day, 5 days a week. Two groups of eight patients each will be conducted in parallel. The complex therapy group will provide additional treatment and place more emphasis on psychological and medical intervention aspects, while patients in the standard group will be advised to exercise on their own during these times. Overall, it is expected that patients will benefit from the structured approach of the programme.

Modules of the MPT programme will be medical lectures and seminars such as social medicine (eg, payment of wages in the event of illness and professional reintegration), information about the MPT concept and special features of MPT and RP, basic information in different lectures (eg, structure and function of the spine, muscles and health-disease conception: bio-psychosocial model) and seminars addressing common questions on topics such as obesity and back pain or correlations of symptoms with stress, sports, nutrition, medications, surgery, lifting and carrying. Another module will consist of medical consultations once a week, alternating between sessions with and without a physiotherapist or psychologist. In addition, regular medical consultations will take place for support in case of abnormalities in the course of the disease and for individual treatment (manual therapy, medication, etc), if needed. Moreover, psychological-pain therapy lectures and seminars will be carried out by psychologists. These will educate patients about pain and stress, mindfulness, problematic behaviour patterns and setting realistic goals. Psychological training sessions will provide practical implementations in form of relaxation techniques, stress management, progressive muscle relaxation and distraction from pain. In addition to individual psychological consultation, regular psychological counselling sessions will be offered. Finally, physiotherapy, occupational therapy, medical training therapy, sports therapy and a module on motivational and volitional aspects will complete the offer.

## AIMS

The objective of the study is to evaluate the benefit of early and intensive relapse prevention in the context of MPT. The goal is to prevent the chronification of back pain and thus reduce the high direct and indirect costs. To answer this question, outcome, process and cost-effectiveness evaluations will be performed. The aim of outcome evaluation is to examine the intervention effect on patient-relevant outcomes. Process evaluation will examine the extent to which the intervention has been carried out and how the process is perceived and evaluated from the perspective of service providers and patients. Indirect and direct costs will be considered in the health economic evaluation.

From this, the following hypotheses can be derived:
► $H_1$: the intervention reduces the days of incapacity to work (outcome evaluation).
► $H_2$: the intervention increases the functional capacity (outcome evaluation).
► $H_3$: the intervention is perceived positively by the participating patients (process evaluation).
► $H_4$: the intervention reduces direct and indirect costs (health economic evaluation).

## METHODS AND ANALYSIS
### Study design

In order to answer the research questions, outcome, process and health economic analyses will be carried out. Different research methods will be used for this purpose. The comparative population will be formed by randomising the groups of participants. The division into the respective groups (intervention group, IG or control group, CG) will be carried out monthly to ensure that the patients of both groups are not able to exchange information with each other. Patients who start an MPT simultaneously in the Rehabilitation Center each form a cluster. These clusters will be randomised to the IG or CG (see figure 2). The evaluator will inform the Rehabilitation Center monthly via email in the third week of the MPT whether the group is an IG or CG. Up to this point, neither the patient nor the treatment staff will know which group the patient belongs to. The recruitment period will be 24 months.

### Outcome evaluation

The results will be derived from a cluster-randomised controlled trial. The evaluation design incorporates an IG as well as a CG and surveys at three different points in time (see figure 2). Both groups will participate in the MPT programme. While IG members will additionally receive relapse prevention modules, the CG will undergo usual care. The primary outcome represents the number of days of incapacity to work within 1 year after the end of MPT. It is assumed that in the present sample, the average number of 37 days off work per year will decrease in the IG by 40% (or 15 days) compared with the CG. One secondary outcome addresses the functional capacity. It is assumed that the functional capacity in the IG will be significantly ($p<0.05$) higher than in the CG. In addition, patient and provider satisfaction with the intervention will be explored, including barriers, hurdles and potentials. Direct and indirect costs are another outcome and will be compared between IG and CG.

### Process evaluation

To evaluate the process quality, we will examine the assignment of participants to the standard and the

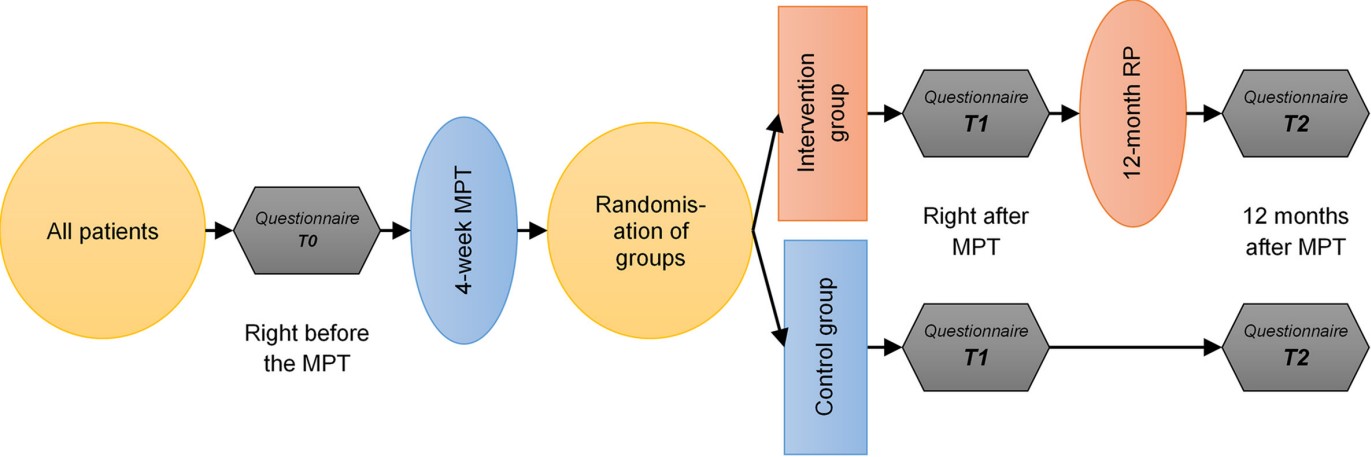

**Figure 2** Study design. MPT, multimodal pain therapy; RP, relapse prophylaxis.

complex programme along with the distribution of the IG to low, medium and high intensity RP. In addition, we will consider usage of the modules (eg, medical training therapy, psychological support and phone consultation) as well as the app utilisation. To examine the quality of the intervention, patients will receive additional questions about their satisfaction with individual MPT and RP modules within the survey. In order to evaluate experiences and satisfaction with the intervention, guided interviews will be conducted. These will be carried out with all physicians, psychologists and physiotherapists involved.

### Health economic analysis

The health economic analysis will examine the costs, cost savings and cost effectiveness of the intervention. Therefore, intervention costs, expenditures of healthcare utilisation, indirect costs due to incapacity to work and the patient's health-related quality of life will be assessed. For the calculation of indirect costs, the days of incapacity to work are multiplied by the average hourly wage in Germany, taking into account whether the individuals work full time or part time. The evaluation will be carried out based on claims data. Incremental cost effectiveness will be analysed by considering direct and indirect costs as well as the health-related quality of life of both groups, IG and CG.

### Measurements

Various instruments will be used in the course of primary data collection, including standardised questionnaires (table 1). Prior to the start of MPT, the IG and CG will receive a questionnaire to assess the status quo at the beginning of the intervention (T0). The questionnaire will be handed out directly at the Rehabilitation Center or sent by mail in individual cases for T0 and after the 4-week MPT programme (T1). It will comprise various items on socio-demographic and socioeconomic parameters together with health-related behaviour (exercise, diet, smoking, etc). Health-related questionnaires will be the Heaviness of Smoking Index,[22] the AUDIT alcohol consumption questionnaire, an effective brief screening test for problem drinking[23] and the Pain Disability Index.[24] In addition, the questionnaire will include items about how often and to what extent the study participants exercise,[25] as well as questions about diet and nutrition.[26] In addition, standardised survey instruments will be included to assess functional capacity (Questionnaire for the diagnosis of functional disability caused by backache, FFbH-R)[27] and health-related quality of life (EQ-5D).[28] A Visual Analogue Scale will be used to assess the patient's health-related quality of life from 0 to 100. IG and CG will receive further questionnaires 12 months post intervention (T2). All questionnaires except for T0 will include questions regarding patient satisfaction and experience

| Target group | Data source | Data type | Instrument | Parameter |
|---|---|---|---|---|
| Patients | Primary data | Quantitative | Questionnaire | Age, gender, sociodemography, satisfaction, functional capacity, health-related quality of life, impairment due to pain, health behaviour, sports activity, nutrition, smoking and alcohol |
| Medical staff and psychologists | Primary data | Qualitative | Interview | Motivation, expectations, experiences, hurdles and barriers, suggestions for improvement and satisfaction with the intervention |
| Patients | Secondary data | Quantitative | Claims data | Days of incapacity to work, region, pharmaceuticals, therapeutic products and outpatient and inpatient care |

**Table 1** Quantitative and qualitative data sources

| Table 2 Inclusion and exclusion criteria | |
|---|---|
| **Inclusion criteria** | |
| Membership | AOK Lower Saxony |
| Region | Northwest of Lower Saxony (see online supplemental figure S1) |
| Sick pay | Entitled |
| Incapacity to work | At least 21 days during the last 6 months |
| Diagnoses | Back pain (ICD: M40–M54, excluding M45, M46 and M49) |
| **Exclusion criteria** | |
| Age | ≥62 years |
| Diagnoses | ▶ Mental and behavioural disorders (ICD: F00–F99; 6 months before) <br> ▶ Malignant neoplasms (ICD: C00–C97; 12 months before incapacity to work) <br> ▶ Injuries (ICD: S00–S99; 6 months before incapacity to work) <br> ▶ Chronic back pain (duration of the incapacity to work within the last 6 months is cumulatively longer than 12 weeks) |

with the intervention. Two weeks after handing out or sending the questionnaire, patients who have not yet responded will receive a reminder to increase the response rate. A second reminder will be sent after an additional 10 days. Reminders and the final questionnaires (T2) for the IG will be sent by the Rehabilitation Center, while those for the CG will be sent by the health insurance fund.

Secondary data will be included in the analyses based on claims data of the regional health insurance fund AOK Lower Saxony. These will include inpatient treatment, rehabilitation services, pharmaceuticals and remedies along with days of incapacity to work. Moreover, sociodemographic factors of influence (eg, age, gender and level of education) together with data regarding health-related quality of life will be included in the analysis.

### Study population
The study population will comprise members of the regional health insurance fund AOK Lower Saxony, residing in the northwest of Lower Saxony (see online supplemental figure S1). The specific inclusion and exclusion criteria are shown in table 2.

### Participant recruitment
Each month, with the exception of December, two MPT courses with eight participants each can be held at the Rehabilitation Center, Bad Zwischenahn. In December, a single course will be held. Over a recruitment period of 24 months, a maximum of 368 individuals could begin an MPT. Potentially eligible individuals will be identified and contacted through health insurance employees. If

they are interested, they will be informed and invited to participate in the programme. A diagnostic day will then be arranged at the Rehabilitation Center (see online supplemental figure S1), where the medical criteria will be reviewed once again.

The risk factors of a possibly low willingness to participate or a high dropout rate of patients were counteracted in advance by the close involvement of the specialised employees of the AOK. For example, the inclusion and exclusion criteria were developed based on their experience. Nevertheless, the development of the case number and the dropout rate will be subject to constant monitoring so that actions can be taken and incentives implemented if necessary.

### Sample size calculation
The case number calculation was carried out for the primary outcome days of incapacity to work. Currently, the described patient group of AOK-insured persons has an average number of 37 days off work (SD: 58) per year. It is assumed that the average number of 37 days off work per year will decrease in the IG by 40% (or 15 days) compared with the CG. This results in an effect size of d=0.3409275. To test with a power of 80% (1–ß=0.8) and a significance level of 5% (α error=0.05), a case number of 137 subjects per group is required for significant proof of the expected effect. The number of cases was calculated using the software G*Power (see online supplemental figureS2).

Taking an intraclass correlation coefficient of 0.01 and a design effect of 1.1 at an average cluster size of 15.3 (22×16+2×8) into account, the required number of cases is 293 subjects, that is, 147 subjects per group.[29–32] Considering a drop-out of 10% for patients who drop out during the MPT, that is, before randomisation, results in a required total case number of 322. The available population covers the required case number (see figure 3 and online supplemental figure S2).

### Statistical analysis
Statistical analysis is performed on the basis of a defined analysis plan. If there are justified deviations from this plan, these will be documented and reported in subsequent publications. Compared with the original plan, the one-sided test with regard to hypothesis $H_4$ was changed to a two-sided test, as this seems reasonable.

### Qualitative analysis
The qualitative guided interviews will be documented by means of audio recordings. The recordings will be transcribed without personal information (eg, name and place of residence). The evaluation of the guided interviews will be based on the qualitative content analysis according to Mayring and carried out with the help of the MAXQDA software.[33]

### Quantitative analysis
All patient-related data will be pseudonymised by assigning a unique identification number. A plausibility

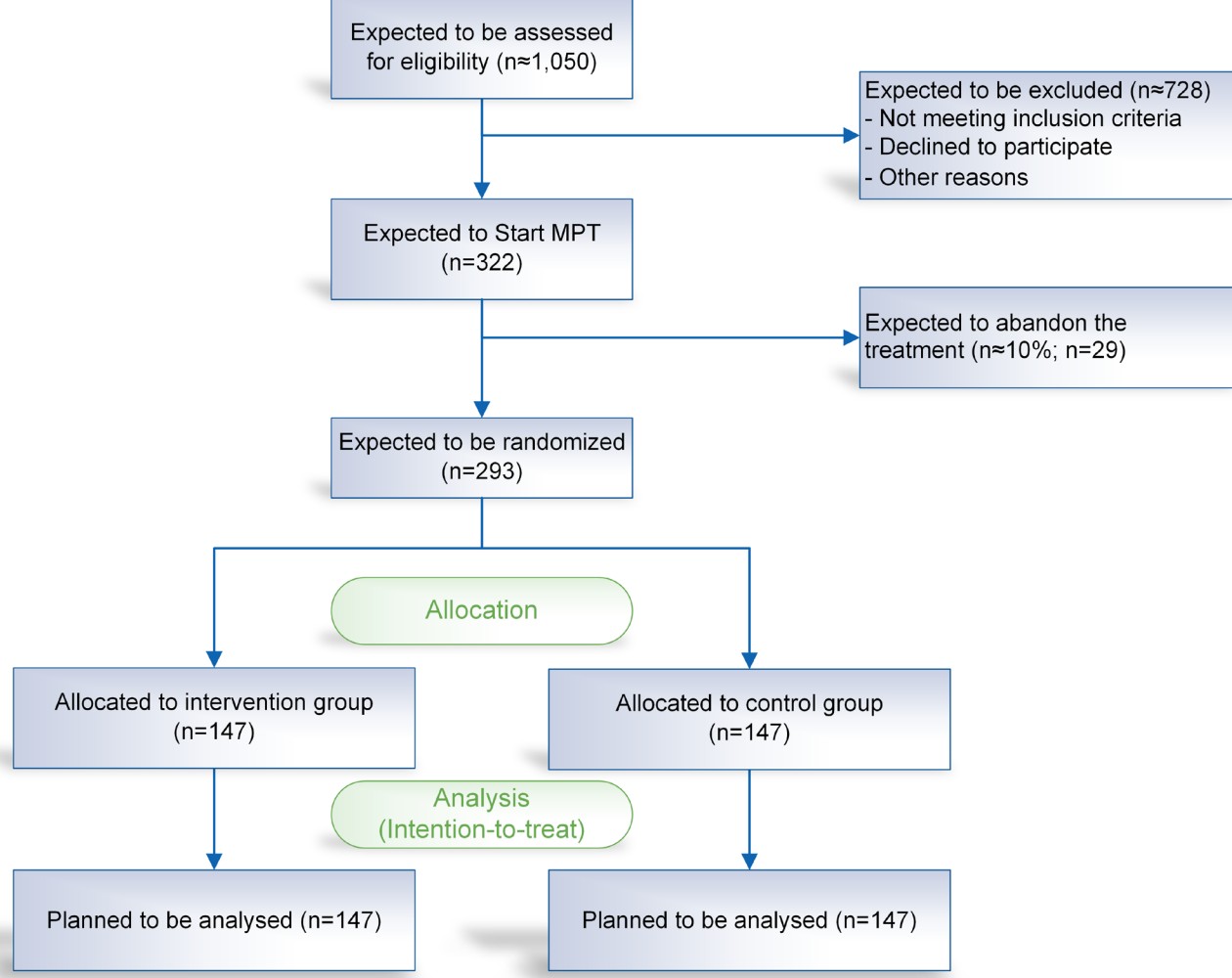

**Figure 3** CONSORT flow diagram. CONSORT, Consolidated Standards of Reporting Trials.

check of the data will be carried out. Initially, descriptive analysis of the data will be performed. Sample characteristics will be analysed with regard to their distribution in the two groups by means of the $\chi^2$ test for binary variables and Mann-Whitney-U test for continuous variables.

### Primary outcome

The data will be analysed by reference to the primary outcome (days of incapacity to work), which is carried out by intention-to-treat analysis. Patients are analysed according to their original group assignment.

To test the primary hypothesis $H_1$ 'The intervention reduces the number of days of incapacity to work', an analysis of covariances with cluster adjustment will be performed, using a significance level of 0.05. Covariables included into the model will be age, gender and comorbidities (eg, depression). In addition, a hypothesis test (two-sided t-test with a significance level of 0.05) will be performed.

### Secondary outcomes

Data for secondary outcome variables will be obtained via questionnaires and claims data. The Markov Chain Monte Carlo approach in SPSS (Version 28) will be used

as imputation method to replace missing values.[34–36] To test the hypothesis $H_2$ 'The intervention increases functional capacity' and $H_4$ 'The intervention reduces direct and indirect costs', hypothesis tests (two-sided t-test with a significance level of 0.05) will be performed. The third hypothesis $H_3$ 'The intervention is perceived positively by the participating patients' will be tested by conducting a pencil–paper questionnaire with the patients. The analyses are carried out using the software programme SPSS (Version 28).

### Patient and public involvement

No patients were involved in the design and conduct of the study, choice of outcome measures and recruitment to the study. The results of the study will be provided to patients at their request.

### ETHICS AND DISSEMINATION

Ethical and scientific standards in the currently valid version will be taken into account when conducting the study, especially the Memorandum on Good Scientific Practice of the German Research Foundation.[37] In

particular, analysis of secondary data will be performed according to the Good Practice of Secondary Data Analysis guidelines, whereas the primary data collection will be conducted in accordance with the Strengthening the Reporting of Observational Studies in Epidemiology statement.[38] In addition, the methods of health economic evaluation in health services research and the Hannover Consensus will be applied.[39] Preliminary to the intervention, a concrete analysis plan is defined. Moreover, the questionnaires developed for the MPT and RP will be pretested and all data will be subjected to a validation process. In addition, participation in the intervention is voluntary for the insured, so there is no disadvantage if people do not want to participate. The applied treatments within the intervention and the potential health problems that may occur are known. Possible adverse events include, for example, pain intensification, pain relapse, muscle strain and other side effects that can occur with any physical activity and specific exercises. The practitioners are prepared and educated for possible adverse events. Experienced practitioners will immediately recognise when adverse events occur during the treatment. Moreover, in the case of an adverse event, they will react appropriately, report and document the adverse event in the electronic patient record. Appropriate action will be taken, such as discontinuation or modification of the treatment.

This study has received approval by the ethics committee of the Hannover Medical School (reference number: 8548_BO_S_2019) and is registered at German Clinical Trials Register DRKS00017654. Each participating patient will sign a contract for participation in the programme, and we will obtain written informed consent. Furthermore, all study participants will be comprehensively informed about the use of the collected data and the purpose and procedure of the project. All transmitted data will be pseudonymised and is accessible only to project members. The delivered data will be stored on a secure and encrypted network drive. Access to the files is restricted to the project staff. No data will be passed on to third parties. Furthermore, the information obtained will be used exclusively for research purposes and no information will be passed on that enables conclusions about individual persons. All data collected and processed will be stored on a secure and encrypted hard disk after the end of the study and deleted 10 years after (by 31 March 2033 at the latest).

The study results will be disseminated in national and international journals and conference presentations. The Innovation Fund (G-BA) will publish the study outcomes in a results report. Additionally, the health insurance fund AOK Lower Saxony will make the results available via its homepage and member journals.

**Author affiliations**
[1]Institute for Epidemiology, Social Medicine and Health Systems Research, Hannover Medical School, Hannover, Germany
[2]Center for Health Economics Research, Hannover, Germany
[3]AOK Niedersachsen, Hannover, Germany
[4]Outpatient Department for Orthopedic Rehabilitation, Rehabilitation Center, Bad Zwischenahn, Germany

**Contributors** KK and CK conceived, designed and obtained funding for the research. The first draft of this manuscript was produced by KK and JS. All authors have provided input to, reviewed, edited and approved the final version. Conceptualisation: KK, CK, WR and GZ. Funding acquisition: CK, MP and NH. Methodology: KK and CK. Project administration: MP. Supervision: CK. Visualisation: JS and KK. Writing—original draft: JS and KK. Writing—review and editing: JS, KK, CK, MP, WR, GZ and NH.

**Funding** This project is funded by the Federal Joint Committee (G-BA), Innovation Fund (funding code: 01NVF18011). The funding company had no role in study design, data collection and analysis, decision to publish or preparation of the manuscript.

**Map disclaimer** The inclusion of any map (including the depiction of any boundaries therein), or of any geographic or locational reference, does not imply the expression of any opinion whatsoever on the part of BMJ concerning the legal status of any country, territory, jurisdiction or area or of its authorities. Any such expression remains solely that of the relevant source and is not endorsed by BMJ. Maps are provided without any warranty of any kind, either express or implied.

**Competing interests** None declared.

**Patient and public involvement** Patients and/or the public were not involved in the design, or conduct, or reporting, or dissemination plans of this research.

**Patient consent for publication** Not applicable.

**Provenance and peer review** Not commissioned; externally peer reviewed.

**ORCID iD**
Julia Schmetsdorf http://orcid.org/0000-0003-4903-0713

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
