## [Reviewer comments · BMJ Open]

ARTICLE DETAILS

TITLE (PROVISIONAL)	Evaluation of a multimodal pain therapy approach with relapse prophylaxis for back pain (MMS-RFP study): study protocol for a cluster randomized controlled trial
AUTHORS	Krueger, Kathrin; Schmetsdorf, Julia Pavlovic, Maja; Runde, Werner; Zechel, Georg; Hemken, Norbert; Krauth, Christian

VERSION 1 – REVIEW

REVIEWER	Shirado, Osamu Fukushima Medical University, Orthopaedic and Spinal Surgery
REVIEW RETURNED	16-Oct-2022

GENERAL COMMENTS	This is a RCT protocol to evaluate a slightly new treatment modality to manage low-back pain (LBP) patients. This RCT would contribute the literature if appropriately planned and successfully performed. However, this reviewer is just wondering if the followings are clearly and practically stated. 1) How do the authors recruit the patients? Who diagnosed if the LBP is “non-specific LBP”? The patients may have lumbar disc herniation, degenerative canal stenosis, discogenic LBP, or facet arthrosis LBP. The subjects’ upper limit of age is 62. How about the lower limit? They should make clear how the LBP is rightly diagnosed, and the subjects are properly selected.2) Is the number of subjects enough to make a clear conclusion? How do the authors determine it?3) Is the study center for the RCT properly built up? The center should be independent of the study team.4) Outcome measures are not clearly mentioned in the protocol. Health-related QOL and disease-specific QOL are not included? What is MCID (minimal clinically important difference) for the QOL measure? What is a rationale to choose it?5) Most importantly, the treatment per se is not clearly described. Who does give it to the patients? And how? Special persons who have a certified technique do this? What is a control group? There is no description on the control group.6) If the adverse events would occur during the trial, who and how responds that? The system to respond it should be clearly made and mentioned. Unfortunately, this reviewer does not think that the article is worthy of be published in The Journal unless they can answer the above-mentioned queries.
---

REVIEWER	Cher, Daniel SI-BONE Inc, Clinical and Regulatory Affairs
REVIEW RETURNED	01-Feb-2023

GENERAL COMMENTS	General comments:  1. The English could be improved by review from a native speaker. 2. The overall design remains unclear in many aspects. Specific comments p. 2, l. 15. "Relapse" should be more carefully defined. p. 2 and throughout: Would write "will be" when describing trial-related activities. p. 2 and throughout: Please avoid using terms like "routine data." p. 3. Good description of study limitations. I think limitations should be described in a separate section in the main text. p. 5 "Chronic recurrent" definition is vague. I suggest making it more specific. p. 5, l. 27: "Chronification" should be defined. p. 5-9. I think the introduction is too long. The introduction should address specific concerns that the trial addresses. p. 9: IG and CG are never defined. I think they mean "interventional group" and "control group". p. 10: The primary endpoint is "days unable to work". It's unclear how this is assessed and whether it is a valid assessment. p. 11: Intensity of the intervention group is described but it's unclear how this is being analyzed or assigned. p. 11: In several areas in the manuscript, the description of how data is collected is vague. Table 1. It's unclear what "experience and satisfaction with the intervention" means if the target group being interviewed is medical staff. The table can be improved. I suggest that eligibility criteria be provided in a table. p. 13: The description of recruitment is too vague. p. 14: I am not convinced that "days unable to work" is normally distributed and so I question the power calculation. p. 15: The primary endpoint analysis will control for variables such as prescription of painkillers, indication-related hospitalization, etc. I'm not sure this is appropriate. Can these be justified? p. 15: Some of the testing is one-sided. I think that there might be a chance that the intervention produces as result in the opposite direction as that expected, and this could be of interest to know. Therefore, 2-sided testing seems more relevant. p. 15: Indirect costs are mentioned. These are not defined. How these are collected is not defined. Two maps of regions in Germany are shown. I'm unclear why two maps are relevant.
---

VERSION 1 – AUTHOR RESPONSE

Reviewer #1

Thank you very much for your review and helpful suggestions. We will provide a point-by-point response in the following. We have taken these on board and, from our point of view, have been able to improve the quality of the paper.

1. How do the authors recruit the patients? Who diagnosed if the LBP is "non-specific LBP"? The patients may have lumbar disc herniation, degenerative canal stenosis, discogenic LBP, or facet

arthrosis LBP. The subjects' upper limit of age is 62. How about the lower limit? They should make clear how the LBP is rightly diagnosed, and the subjects are properly selected.

As stated on page 10 above, employees of the health insurance fund identify and contact the potentially eligible insured (ICD M40-M54, excluding M45, M46 and M49) and refer them to the rehabilitation center. A comprehensive initial examination is then performed by the orthopedics, physiotherapists, and psychologists at the rehabilitation center prior to the start of multimodal pain therapy. They ensure that the inclusion criteria are met (see p. 16). We have added the relevant information (p. 10).

As the above ICD diagnoses make clear, individuals with problems of the entire spine were potentially eligible for inclusion. Therefore, we corrected this and replaced the term LBP with back pain. Thank you for pointing this out.

In our understanding, specific back pain has an attributable somatic cause (e.g., the so-called red flags: fracture, spondylitis, tumor, nucleus pulposus prolapse with radiculopathy), which consequently requires specific therapy, hence no Multimodal Pain Therapy (MPT).

However, non-specific back pain may occur with structural changes. In our approach, back pain is non-specific in the sense that it does not allow a clear conclusion on a structural pathology, but occurs in the same form with several possible causes (e.g., osteochondrosis, spondylolisthesis, nucleus pulposus prolapse without radiculopathy). In addition, there is no conservative specific therapy to causally affect the structural changes. During the multimodal assessment, it was ensured that the included back pain did not require specific therapy (e.g., herniated disc with existing radiculopathy), but could be treated by multimodal therapy within the MPT. However, to avoid misunderstandings, we have removed the term non-specific.

The study population comprises members of the regional health insurance fund who are entitled to receive sick pay and have been incapacitated for work for at least 21 days during the last six months due to a diagnosis of back pain (see p. 16). Thus, there was no lower age limit, but persons could be included from the beginning of employment.

2. Is the number of subjects enough to make a clear conclusion? How do the authors determine it?

We carried out a sample size calculation (see page 17 ff.) to determine the sample size that is required to test our hypotheses. The case number calculation is carried out for the primary outcome days of incapacity to work.

3. Is the study center for the RCT properly built up? The center should be independent of the study team.

Thank you for pointing this out. The rehabilitation center operates independently of the study team. The study team has no influence on the work of the physicians, psychologists, or other disciplines such as physiotherapists and others conducting the treatment. The study team consists of researchers from an institute at the Hannover Medical School, and the study center is a completely independent rehabilitation center about 200 km away.

We added the following sentence to the revised manuscript. "The rehabilitation center is completely independent of the evaluating institution." (see page 8)

4. Outcome measures are not clearly mentioned in the protocol. Health-related QOL and disease-specific QOL are not included? What is MCID (minimal clinically important difference) for the QOL measure? What is a rationale to choose it?

We have added the relevant information and outcome measures (see p. 14 ff.). Health-related QoL will be assessed by EQ-5D (five levels, 5L) and as a proxy for disease-specific QoL the Pain Disability Index (PDI) and the Hannover Functional Ability Questionnaire for measuring back pain-related functional limitations (German title: Funktionsfragebogen Hannover Rücken (FFbH-R)) are used (see p. 14).

Participants indicate their current health status in five health domains (pain, mobility, usual activities, self-care, anxiety/depression) using a five-point scale (Version EQ-5D-5L). The EQ-5D is a standardized and widely used preference-based quality-of-life measurement tool.

The MCID is not examined in this study. Soer et al. analyzed clinimetric properties of the EuroQoL-5D in patients with chronic low back pain and found an MCID of 0.03 points for the categorical scales of the EQ-5D and 10.5 points for the EQ-5D visual analog scale.

5. Most importantly, the treatment per se is not clearly described. Who does give it to the patients? And how? Special persons who have a certified technique do this? What is a control group? There is no description on the control group.

The treatment description follows the sentence "The intervention examined in this study..." on the following pages (8 to 11). As different expert groups are involved in the treatment, they apply their expert knowledge as well as certified techniques in which they are trained and which can be applied specifically to the patients included. We revised the paper with regard to your comment and sought to specify who will provide patient treatment during the trial and what exercises will be performed (see page 10 and 11).

While IG members additionally receive relapse prevention modules, the CG undergoes usual care (see p. 12). Moreover, we added further information (see Abstract): „IG and CG receive MPT, after randomization IG receives RFP and CG receives no further therapy or support as part of the trial.”

6. If the adverse events would occur during the trial, who and how responds that? The system to respond it should be clearly made and mentioned.

Thank you for pointing this out. Adverse events were not anticipated to the extent that they might occur, for example, in the context of a clinical trial. Since the MPT and RFP include sports exercises and psychological counselling, adverse events cannot be assumed here. For this reason, adverse events were not addressed. Nevertheless, rehabilitation center staff could intervene in the event of an adverse event and the participants were able to contact the orthopedics, physiotherapists, and psychologists directly (during MPT) and indirectly at any time. In addition, surgery would be recommended if necessary or the intervention would be discontinued if the patient's condition worsened.

Reviewer #2

Thank you for your review and helpful suggestions. Please find our adaptations and responses below.

1. p. 2, l. 15. "Relapse" should be more carefully defined.

We have specified this in the manuscript: "Relapse refers to the recurrence of back pain after temporary improvement." (see page 6)

2. p. 2 and throughout: Would write "will be" when describing trial-related activities.

We have revised the paper with regard to your suggestion.

3. p. 2 and throughout: Please avoid using terms like "routine data."

Thank you for the comment. We have replaced the word routine data with claims data throughout the manuscript.

4. p. 3. Good description of study limitations. I think limitations should be described in a separate section in the main text.

We have followed the submission guideline of BMJ Open : "Please include a 'Strengths and limitations of this study' section after the abstract." Thank you for pointing this out.

5. p. 5 "Chronic recurrent" definition is vague. I suggest making it more specific.

Thank you for the comment. We specified this aspect: "The term recurrent describes pain episodes that reoccur after a symptom-free period of at least six months." (see page 8 above)

6. p. 5, l. 27: "Chronification" should be defined.

We added this information by adding the phrase "Chronification moreover, describes the transition from temporary to permanent, or chronic, back pain." (see page 8 above)

7. p. 5-9. I think the introduction is too long. The introduction should address specific concerns that the trial addresses.

We have shortened the introduction and focused on the information relevant to the study.

8. p. 9: IG and CG are never defined. I think they mean "interventional group" and "control group".

Yes, that is correct, thank you. We have added this to the manuscript (see page 12).

9. p. 10: The primary endpoint is "days unable to work". It's unclear how this is assessed and whether it is a valid assessment.

The data basis for "days unable to work" are the days of incapacity for work, which are available via the claims data of the health insurance fund. The information on days of incapacity for work is based on incapacity certificates from the treating physician, which also contain the ICD diagnosis determined as the cause for the incapacity for work. These data are transmitted to the health insurance fund. The claims data are provided for all participants. However, it should be taken into account that incapacity certificates are usually only provided from the third day of incapacity for work, so that there could be an underestimation here, which, however, applies across the board to all participants in both the intervention and the control group. Furthermore, for back pain, the duration of incapacity to work is generally expected to be greater than two days, and since only days of incapacity to work for back pain (using ICD codes) are used for the primary outcome, the impact is likely to be small. We have added this point as a limitation: "As a limitation for the primary outcome, it should be taken into account that the days of incapacity to work are based on incapacity certificates issued by the treating physician, which are usually only submitted to the health insurance funds for incapacity to work lasting three days or longer. However, this applies to both the intervention and the control group. Furthermore, for back pain, the duration of incapacity to work is generally expected to be greater than

two days, and since only days of incapacity to work for back pain (using ICD codes) are used for the primary outcome, the impact is likely to be small.” (see page 4 ff.)

10. p. 11: Intensity of the intervention group is described but it's unclear how this is being analyzed or assigned.

Intensity is assigned by physicians based on the patient's needs and will be considered as a potential influencing factor in further analyses. It is not among the primary and secondary endpoints but we will check if there is a correlation with the severity of the back pain at the beginning of the intervention.

11. p. 11: In several areas in the manuscript, the description of how data is collected is vague.

We have revised the relevant parts (see p. 4 Strengths and limitations of this study, p. 13 ff. and p. 19).

12. Table 1. It's unclear what "experience and satisfaction with the intervention" means if the target group being interviewed is medical staff. The table can be improved.

The focus here is on process evaluation and the question of how those involved in the implementation of the intervention (i.e., the treating physicians, physiotherapists, psychologists and occupational therapist) perceive and evaluate the intervention. Since the project is an evaluation of a new form of care, it is also important in the process evaluation to find out how processes could be optimized, if necessary. To this end, the motivation, expectations, experiences, hurdles, and barriers as well as suggestions for improvement and satisfaction are recorded. We revised the table (see p. 15).

13. I suggest that eligibility criteria be provided in a table.

We added a table presenting the eligibility criteria more clearly (see p. 16).

14. p. 13: The description of recruitment is too vague.

We specified this aspect in the manuscript: “Potentially eligible individuals are identified and contacted through health insurance employees. If they are interested, they are informed and invited to participate in the program. A diagnostic day is then arranged at the rehabilitation center, where the medical criteria are reviewed once again.” (see page 17 above)

15. p. 14: I am not convinced that "days unable to work" is normally distributed and so I question the power calculation.

We will check whether the data on incapacity to work are normally distributed. While the normal distribution assumption is theoretically important for the unpaired t-test, numerous studies have shown in practice that the unpaired t-test is relatively robust to violations of the normal distribution assumption. Since we cannot adjust the number of cases now, in the course of the project, we would refrain from an updated case number calculation and use the t-test. Of course, if there is no normal distribution, additional non-parametric tests can be carried out. We have performed case number calculations in G*Power for non-parametric tests with different distributions. The number of cases already calculated (n=322) would cover the number of cases for a Wilcoxon-Mann-Whitney test (two groups) with Laplace distribution (n=215) or logistic distribution (n=295).

16. p. 15: The primary endpoint analysis will control for variables such as prescription of painkillers, indication-related hospitalization, etc. I'm not sure this is appropriate. Can these be justified?

Upon further inspection, we decided to cut the sentences from the manuscript. (see p. 19 above).

17. p. 15: Some of the testing is one-sided. I think that there might be a chance that the intervention produces as result in the opposite direction as that expected, and this could be of interest to know. Therefore, 2-sided testing seems more relevant.

That is an important hint, thank you. We will take it up and test the hypotheses b (The intervention increases functional capacity.) and d (The intervention reduces direct and indirect costs.) on both sides. We have changed this point in the manuscript (see p. 19).

18. p. 15: Indirect costs are mentioned. These are not defined. How these are collected is not defined.

Thank you very much for your comment. The basis for the calculation of the indirect costs are the days of incapacity to work, which are available via the claims data of the health insurance fund. Taking into account whether the insured work full-time or part-time, the average hourly wage is used to determine the amount of indirect costs. The amount of the average hourly wage in Germany is available from the Federal Statistical Office (Destatis). We have specified this point in the manuscript: "For the calculation of indirect costs, the days of incapacity to work are multiplied by the average hourly wage in Germany, taking into account whether the individuals work full-time or part-time." (see page 13)

19. Two maps of regions in Germany are shown. I'm unclear why two maps are relevant.

One figure shows where in Germany the area of the eligible participants is located. The second figure shows in more detail where in the state of Lower Saxony the area is located. But we have removed one figure to avoid confusion about the location of the study region (see Figure 3). Thank you for the hint.

VERSION 2 – REVIEW

REVIEWER	Shirado, Osamu Fukushima Medical University, Orthopaedic and Spinal Surgery
REVIEW RETURNED	21-Apr-2023

GENERAL COMMENTS	This reviewer does not think that the author's answers # 6 is enough to be accepted for publication. Any studies including any interventions should have the system that can prepare adverse events.
--

VERSION 2 – AUTHOR RESPONSE

Reviewer #1

1. Any studies including any interventions should have the system that can prepare adverse events.

Thank you for your helpful suggestion. In consultation with the treatment providers, we again considered and elaborated the point of adverse events related to the intervention in greater depth.

To determine the eligibility of study participants, a comprehensive multiprofessional assessment will be conducted prior to intervention. The applied treatments within the intervention and the problems that may occur are known. Possible adverse events include f. ex. pain intensification, pain relapse, muscle strain and other side effects that can occur with physical activity and specific exercises. In individual cases, overexertion of the cardiovascular system and pain in the adjacent joints may occur, as well as restlessness as a negative side effect of PMR (progressive muscle relaxation). In addition, the psychological specialists will closely monitor known side effects of psychological interventions.

The practitioners are prepared and educated for possible adverse events. Experienced practitioners will immediately recognize when adverse events occur during the treatment. Moreover, in the case of an adverse event, they will react appropriately, report and document the adverse event in the electronic patient record. Appropriate action will be taken, such as discontinuation or modification of the treatment.

We have added the relevant information to the revised manuscript (see page 20).